# Mental Health Training, Attitudes toward Support, and Screening Positive for Mental Disorders among Canadian Coast Guard and Conservation and Protection Officers

**DOI:** 10.3390/ijerph192315734

**Published:** 2022-11-26

**Authors:** Katie L. Andrews, Laleh Jamshidi, Jolan Nisbet, Taylor A. Teckchandani, Jill A. B. Price, Rosemary Ricciardelli, Gregory S. Anderson, R. Nicholas Carleton

**Affiliations:** 1Canadian Institute of Public Safety Research and Treatment (CIPSRT), University of Regina, Regina, SK S4S 0A2, Canada; 2Fisheries and Marine Institute, Memorial University of Newfoundland, St. John’s, NL A1C 5R3, Canada; 3Faculty of Science, Thompson Rivers University, Kamloops, BC V2C 0C8, Canada

**Keywords:** critical incident stress debriefing (CISD), critical incident stress management (CISM), mental health first aid, peer support, road to mental readiness (R2MR), posttraumatic stress injuries (PTSIS), occupational stress injuries (OSIS), public safety personnel (PSP)

## Abstract

Public Safety Personnel (PSP) including members of the Canadian Coast Guard (CCG) and Conservation and Protection (C&P) officers, are regularly exposed to potentially psychologically traumatic events (PPTEs) and other occupational stressors. Several mental health training programs (e.g., critical incident stress management [CISM], critical incident stress debriefing [CISD], peer support, mental health first aid, Road to Mental Readiness [R2MR]) exist as efforts to minimize the impact of exposures. To help inform on the impact of several categories of mental health training programs (i.e., CISM, CISD, mental health first aid, Peer Support, R2MR) for improving attitudes toward support and willingness to access supports among CCG and C&P officers, the current study assessed CCG and C&P Officers perceptions of access to professional (i.e., physicians, psychologists, psychiatrists, employee assistance programs, chaplains) and non-professional (i.e., spouse, friends, colleagues, leadership) support, and associations between training and mental health. Participants (*n* = 341; 58.4% male) completed an online survey assessing perceptions of support, experience with mental health training and symptoms of mental health disorders. CCG and C&P Officers reported access to professional and non-professional support; however, most indicated they would first access a spouse (73.8%), a friend (64.7%), or a physician (52.9%). Many participants would never, or only as a last resort, access other professional supports (24.0% to 47.9%), a CCG or C&P colleague (67.5%), or their leadership (75.7%). Participants who received any mental health training reported a lower prevalence of positive screens for all mental health disorders compared to those who did not received training; but no statistically significant associations were observed between mental health training categories and decreased odds for screening positive for mental disorders. The current results suggest that the mental health training categories yield comparable results; nevertheless, further research is needed to assess the shared and unique content across each training program. The results highlight the need to increase willingness to access professional and non-professional support among CCG and C&P Officers. Revisions to training programs for leadership and colleagues to reduce stigma around mental health challenges and support for PSP spouses, friends, and physicians may be beneficial.

## 1. Introduction

Regular exposure to potentially psychologically traumatic events (PPTEs) such as exposure to threatened or actual physical assaults, serious injury, fires, or explosions [1] is expected during occupational activities of Public Safety Personnel (PSP) [2]. PSP include persons working within the Canadian Coast Guard (CCG) and Conservation and Protection Services (C&P) [3]. CCG and C&P officers have previously reported exposure to an average of eight different PPTE types, with each type having been experienced 10 or more times by up to 78.9% of respondents [3]. High exposure frequencies have also been observed among other Canadian PSP (i.e., municipal/provincial police, firefighters, paramedics, Royal Canadian Mounted Police, correctional workers, and dispatchers) [2]. Among CCG and C&P, PPTE exposures have been reported to be associated with increased odds of screening positive for mental health disorders [3] and suicidal behaviors [4]. Approximately 42.0% of CCG and C&P officers screened positive for one or more mental health disorders [5] as well as high proportions reporting lifetime suicidal ideation (25.7%), planning (10.9%), and attempts (5.5%) [4]. A high prevalence of positive screens for mental health disorders [6] and suicidal behaviors [7] have also been observed among other Canadian PSP. The evidence suggests that the frequent exposure to PPTEs experienced by PSP are problematic and, at least in part, related to the mental health challenges observed among PSP.

Mental health challenges represent a significant concern for PSP organizations and leadership. Several strategies and programs have been developed and deployed among PSP to minimize and manage the impact of PPTEs. The available programs can be categorized into proactive and reactive programs. Proactive programs (i.e., mindfulness, psychoeducation, resilience promotion, psychophysiology, cognitive behavioural therapy) are intended to be implemented before PPTE exposures to “prevent” or mitigate the development of posttraumatic stress injuries (PTSI) rather than treat them [8]. Road to Mental Readiness (R2MR) is an example of one type of proactive program designed to provide evidence-based psychoeducation about stress, trauma, and coping to build resilience, reduce stigma and barriers to care, encourage early access to care, and provide tools to manage mental health disorder symptoms [9]. The content is generally presented as a 4 h (for frontline employees) or 8 h (for supervisors) classroom based educational program.

Reactive programs (i.e., Peer Support, Mental Health First Aid, critical incident stress management [CISM], critical incident stress debriefing [CISD]) are generally implemented to manage and reduce the impact of PPTE exposures. CISM provides detailed, integrative, and multi-phase peer support that: (1) incorporates specific tools tailored for psychological injury and stress (i.e., resistance, resilience, recovery); (2) emphasizes the value in peer relationships to reconnect individuals to their adaptive coping strategies; and (3) fosters group cohesion, performance, and social connections [10]. CISD was developed specifically for PSP and is often used to manage an individual’s acute stress response immediately following a PPTE exposure [11]. CISD is typically intended to assist and support in the context of work-related stressors [12,13]. Both programs focus on awareness and encourage PSP to engage emotionally, recognizing that an individual’s coping mechanisms may be overwhelmed after a PPTE exposure [13]. Peer support provides opportunities for PSP to (1) talk to individuals who are familiar with and understand the unique demands of PSP work; (2) access relative formal mental health resources; and (3) feel more comfortable with a peer than they might with a registered mental health care provider [14]. Mental Health First Aid is designed to increase the trainees’ knowledge of mental health concerns, decrease stigmatizing attitudes toward people with mental health disorders and increase confidence and helping behaviors when mental health concerns are recognized in others [15,16]. Mental Health First Aid is intended to help participants recognize common mental health disorders, increase awareness of treatment options and self-help strategies, and develop skills to use in a mental health crisis [17].

Despite the availability of training programs, there is limited research regarding program effectiveness. There have been few evaluations of both proactive and reactive mental health training programs for PSP [10,13,17,18,19]. A recent systematic review of the available literature observed high variability in study design, target audience, duration of training, time of interventions, outcomes measured and timing of follow-up across the programs and associated evaluations [19]. Accordingly, comparing the effectiveness of the programs is extremely difficult and quality assessments of the impact of such programs on the mental health of PSP is rarely available [19]. The same review reported that CISM and CISD effectiveness has not been robustly assessed due to inconsistencies across studies. The conclusions were consistent with a previous literature review conducted in 2016 [13]. In a more current study [10], Canadian firefighters and paramedics perceived CISM as a beneficial and valuable tool providing skills and coping strategies. CISM was also reported to offer some mental health benefits for symptoms of Alcohol Use Disorder (AUD) and Generalized Anxiety Disorder (GAD) when delivered with high fidelity [10]. Another meta-analysis examining the effectiveness of Mental Health First Aid, observed moderate improvements in mental health knowledge and confidence of trainees to help those in need [17]. The results were inconclusive for recipients of help from the trainees. 

The studies included in the systematic review examining R2MR reported favorable results including some improvements in mental health outcomes and stigma [19]. Similarly, in a meta-analysis of the available R2MR literature, R2MR for PSP was observed to reduce stigmatizing attitudes towards those with mental health disorders and increase resiliency skills, although effects were time limited with rapid skill decay [18]. These results were consistent with those of a study of paramedic students [20]. In a recent review of proactive mental health training programs, resilience promotion and multimodal programs that combine a variety of therapeutic and skill building approaches were concluded to produce modest time-limited reductions in symptoms of general psychological health, depression, burnout, stress, posttraumatic stress disorder (PTSD), and anxiety, as well as promoting well-being, adaptive coping and resilience [19]. The available evidence suggests both proactive and reactive programs may be effective at improving mental health knowledge and reducing stigma, however a great deal of heterogeneity was observed across studies and the evaluated programs leading to substantial barriers to evaluating program effectiveness [13,19] With limited evidence and substantial barriers to evaluating and comparing all the available programs, Canadian PSP leaders are faced with ambiguity regarding which program(s) to implement to manage and mitigate the impact of PPTEs and other occupational stressors. 

Research is also limited regarding the impact of training on improving attitudes toward support and willingness to access supports among PSP. Previous research including a diverse sample of Canadian PSP [21], reported most had access to both professional and non-professional supports; however, PSP who received any mental health training reported modestly higher levels of perceived access, indicated higher willingness to access all types of support, and reported lower positive screens for any mental health disorder than those with no training [21]. The evidence suggests that any mental health training (proactive or reactive) may be beneficial for improving mental health and willingness to access mental health supports [19,21]. Nevertheless, the previous research has not included CCG and C&P officers. Identifying programs effective in improving CCG and C&P attitudes toward support, willingness to access support, and associated decreases in positive screens for mental health disorders will provide PSP organizational leadership with information about which programs to prioritize for their members. 

The current study was designed based on previous research [21] to help inform on the impact of several categories of mental health training programs (i.e., CISM, CISD, Mental Health First Aid, Peer Support, R2MR) for improving attitudes toward support and willingness to access supports among CCG and C&P officers. Specifically, the current study assesses CCG and C&P Officers attitudes toward accessing support from professional (i.e., physicians, psychologists, psychiatrists, employee assistance programs, chaplains) and non-professional sources (i.e., spouse, friends, colleagues, leadership) based on participation in different mental health training programs and screening positive for one or more mental health disorders. The current study will inform decisions about which specific mental health training programs to implement to improve attitudes towards supports and willingness to access both professional and non-professional supports. Based on previous research with Canadian PSP [21], training program participation was expected to be associated with higher awareness of available support, willingness to access support, and decreased odds of screening positive for mental health disorders; however, there were no specific directional hypotheses about the different training programs.

## 2. Materials and Methods

### 2.1. Procedure

Data were collected using a web-based self-report survey available in both English and French. The study was approved by the University of Regina Institutional Research Ethics Board (REB# 2021-003). The survey was based on a set of validated measures used in a previous study of PSP [2,6,7,21,22], but collaboratively redesigned by members of the research team and the CCG and DFO team to ensure relevant variables were included. The survey was promoted and distributed by the CCG and DFO to member unions via emails, social media posts, and a video encouraging participation. The survey was available from 1 February 2021 to 31 January 2022. At the start of the survey participants selected their preferred language (i.e., English or French) in which to complete the survey and were presented with the study information and an informed consent form. Participation was anonymous and voluntary, and each respondent was provided a randomly generated unique code which allowed for repeated survey access to complete the survey over multiple sessions. The current study focused specifically on self-reported perceptions of access to professional and non-professional support, mental health training experience, and positive screens for mental health disorders based on several well-established measures assessing mental disorder symptoms.

### 2.2. Data and Sample

Participants were CCG/C&P members (*n* = 412) (67.5% CCG members and 26.0% C&P members). Responses from 561 CCG/DFO members were initially collected, but only data from respondents who completed at least 30% of the survey were retained. The final sample was a total of 412 respondents. For the current study, data from respondents who completed the sections on mental health training and attitudes toward mental health were included in the current analyses and results. Participants were mainly male (56.1%), identifying as men (55.3%), white (i.e., Caucasian) (82.8%), and aged 30 to 39 years old (26.9%) or 40 to 49 years old (26.5%) (see Table 1). Participants were mostly married or in common-law relationships (i.e., living with a person in a conjugal relationship for 12 continuous months) (63.8%), with a college (37.4%) or a university (31.3%) degree, residing in British Columbia (53.2%), with no previous experience as either PSP or in the Canadian Armed Forces (CAF) (67.5%).

### 2.3. Mental Health Training

Questions assessing mental health training were based on a set of questions used in a previous study of Canadian PSP [21]. Participants were asked to indicate all of the different categories of mental health training they received in their CCG and C&P role. Formal mental health training options included: (a) Critical Incident Stress Management (CISM); (b) Critical Incident Stress Debriefing (CISD); (c) Mental Health First Aid; (d) Peer Support; and (e) Road to Mental Readiness (R2MR). Response options were not mutually exclusive and, as such, respondents could indicate all that apply. An “any mental health training” variable was comprised of respondents who received one or more of the first five categories of mental health training. The “other mental health training” category had a small sample size and was not included as an additional individual training type category but was included within the “any mental health training” variable. Following a positive response, participants were asked to specify what training they had received and whether the training was perceived as helpful.

### 2.4. Attitudes toward Mental Health Support

Attitudes toward accessing both professional and non-professional mental health support were assessed with a set of questions used in a previous study of Canadian PSP [21]. The following stem question was asked: “Which of the following potential support resources do you feel you can access if you need help managing your mental health?” Professional mental health support included: (a) employee assistant program; (b) physician; (c) psychiatrist; (d) psychologist; and (e) religious or spiritual leader. Non-professional mental health support included: (a) a colleague; (b) leadership; (c) friend; and (d) spouse. Response options included: (1) I can and would access as an early resource; (2) I can access, but only as a last resort; (3) I can access but would never; (4) I do not have access, but I would access as an early resource; (5) I do not have access but would access only as a last resort; (6) I do not have access but would never access, and (7) I do not know if I have access. Response options 2 and 3 and options 5 and 6 were collapsed to create a 5-point scale due to small sample size in each category, therein supporting robust solutions, protecting against potential confidentiality concerns, and allowing for comparison with previously collected PSP data [21].

### 2.5. Mental Health Disorder Screens

Mental health disorder symptoms were assessed by self-report using the Posttraumatic Stress Disorder (PTSD) Checklist for DSM-5 (PCL-5) [23,24]; the 9-item Patient Health Questionnaire (PHQ-9) [25] indexing Major Depressive Disorder (MDD) symptoms; the Panic Disorder Symptoms Severity scale, Self-Report (PDSS-SR) [26] indexing panic disorder (PD) symptoms; the 7-item Generalized Anxiety Disorder scale (GAD-7) [27] indexing GAD symptoms; the Social Interaction Phobia Scale (SIPS) [28] indexing Social Anxiety Disorder (SAD) symptoms; and the Alcohol Use Disorders Identification Test (AUDIT) [29] indexing AUD symptoms. Participants reported their behaviors over the last year for the AUDIT, the past month for the PCL-5, the past 14 days for the PHQ-9 and GAD-7, and the past 7 days for the PDSS-SR. There is no specific time window used for SIPS. For the PCL-5, a positive screen required participants to report exposure to at least one item from the Life Events Checklist for DSM-5 (LEC-5) [30], meet minimum DSM-5 criteria for each PTSD symptom cluster subscale [31] (e.g., intrusions, avoidance, negative alterations in cognitions and mood, and alterations in arousal and reactivity), and exceed the clinical cut-off of >32 [24]. A positive screen required the PHQ-9 total score to be >9 [6], the PDSS-SR total score to be >7 [26], the GAD total score to be >9 [27], the SIPS total score to be >20 [28], and the AUDIT total score to be >15 [32].

### 2.6. Statistical Analyses

Cross-tabulations were performed to determine the prevalence of attitudes toward each of the professional and non-professional mental health supports for each mental health training category (i.e., CISM, CISD, Mental Health First Aid, Peer Support, R2MR), as well as any training and no training. Prevalence for the “other” mental health training category was not reported in the current study due to small sample size. Cross-tabulations were conducted to determine the prevalence of positive screens for mental health disorders for each mental health training category. Logistic regression models were used to examine the association between mental health training categories and each mental disorder screen. Covariates included sex, gender, age, education, ethnicity, marital status, province of work, and job category. Computed logistic regressions models included: (1) unadjusted regression model (OR); (2) adjusted model 1 adjusted for sex, gender age, education, ethnicity, marital status, province, and job category covariates (AOR1); and (3) adjusted model 2 adjusted for all covariates from AOR1 in addition to all categories of mental health training (i.e., CISM, CISD, Mental Health First Aid, Peer Support, R2MR) (AOR2). Cross-tabulations were conducted to determine the prevalence of attitudes toward the effectiveness of mental health training of each mental health training category. 

## 3. Results

The prevalence of attitudes toward professional support for each mental health training category, as well as any training and no training are presented in Table 1. For all mental health training categories, respondents reported the highest prevalence for accessing physicians indicating “I can and would access as an early resource” if they required support managing their mental health (prevalence ranged from 48.0% to 58.2%), and the lowest prevalence for accessing chaplains (prevalence ranged from 9.0% to 21.1%). A higher prevalence of respondents indicated “I can access but would never/only as a last resort” for all professional mental health supports except physicians compared to the “can and would access as an early resource” response option. Few respondents indicated “I don’t know if I have access” for psychologists, psychiatrists, and employee assistance program (prevalence ranged between 3.6% to 16.7%); whereas, between 15.8% and 26.0% indicated “I don’t know if I have access” for chaplains. For several professional mental health support categories (i.e., physician, employee assistance, chaplain), a lower proportion of participants indicating “I can and would access” was observed for the “no mental health training” group compared to the “any mental health training” group.

Prevalence of attitudes toward non-professional mental health support for each mental health training category are presented in Table 2. Spousal support was the most common non-professional support indicated “I can and would access as an early resource” (prevalence ranged from 63.3% to 75.7%), followed by friends (prevalence ranged from 63.9% to 71.1%). CCG and C&P officers indicated low prevalence of accessing leadership as an early resource (prevalence ranged from 11.2% to 13.3%) for help managing their mental health. Very few respondents in the any training category indicated “I don’t have access, but would access” for support from a friend, CCG or C&P colleague, or CCG or C&P leadership, whereas a higher prevalence was indicated for spousal support (prevalence range from 5.4% to 8.8%). 

The prevalence of screening positive for a mental health disorder associated with having received each mental health training category is presented in Table 3. Participants who received any mental health training reported a smaller prevalence of screening positive for all mental health disorders compared to those individuals who did not receive training (prevalence range from 7.5% to 22.4% for any training compared to 8.5% to 31.4% for no training). 

The results assessing the associations between mental health training categories and positive screens for mental health disorders are presented in Table 4. CISM training was statistically significantly related to decreased odds of screening positive for GAD (odds ratio [OR] = 0.33, 95% CI = 0.13–0.84). CISD training was statistically significantly associated with decreased odds of screening positive for MDD (OR = 0.28, 95% CI = 0.09–0.96). None of the mental health training categories were statistically significantly associated with changes in odds of screening positive for PTSD, SAD, PD, and AUD. After adjusting for sociodemographic covariates (i.e., sex, age, education, ethnicity, marital status, province, job category), only peer support was statistically significantly associated with decreased odds of screening positive for MDD (adjusted odds ratio [AOR] = 0.34, 95% CI = 0.12–0.97). After adjusting for other mental health training categories and sociodemographic covariates, none of the mental health training categories were statistically significantly related to changes in odds of screening positive for all mental health disorders. 

Prevalence of attitudes toward the effectiveness of each mental health training category are presented in Table 5. Participants reported CISD as most effective for improving mental health (66.7%), reducing stigma (66.7%), increasing their knowledge about mental health (81.8%), and helping them to respond to members of the public with mental health problems (69.7%). Peer support was reported to have the highest impact on improving the mental health of participants’ team members (64.1%) and reducing mental health injuries (46.2%). Participants across all mental health training categories reported training as least effective for mitigating occupational stress injuries (range from 29.3% to 46.2%) and most effective for increasing their knowledge about mental health (range from 71.7% to 81.8%). Participants who received any mental health training reported training as effective for all objectives except mitigating occupational stress injuries. 

## 4. Discussion

The current study assessed CCG and C&P attitudes towards accessing support from professional (i.e., physicians, psychologists, psychiatrists, employee assistance programs, chaplains) and non-professional (i.e., spouse, friend, colleagues, leadership) sources based on participating in different training program categories (i.e., CISM, CISD, Mental Health First Aid, peer support, R2MR). The current results inform decisions about which specific mental health training programs to implement to improve attitudes towards supports and willingness to access both professional and non-professional supports. Based on previous research [21], participation in any training program was expected to be associated with higher willingness to access support and lower odds of screening positive for mental health disorders; but there were no specific expectations about differences between training programs. CCG and C&P officers reported having access to all types of professional and non-professional supports. Participants who received any training program indicated a higher willingness to access professional and non-professional supports as an early resource for mental health challenges compared to those with no training; however, willingness to access support varied based on the type of professional or non-professional. The observed results were consistent across all types of training programs. A lower prevalence of positive screens for all mental disorders was observed among those with any training compared to those with no training. CISM, CISD, and Peer Support were statistically significantly associated with decreased odds of screening positive for some mental health disorders; but after adjusting for covariates and other mental health training categories, none of the mental health training categories were statistically significantly associated with changes in positive screens for all mental health disorders. 

The current results indicate that, even without any mental health training, most CCG and C&P officers believe they have access to professional supports, but willingness varied based on the type of profession. CCG and C&P with no mental health training were most willing to access support from a physician as an early resource (49.2%) and would never or only as a last resort access support from a psychologist (42.5%), a psychiatrist (41.5%), a chaplain (27.1%), and their employee assistance program (65.3%). The largest proportion of CCG and C&P (65.3%) reported they would never or only as a last resort access their employee assistance program. The employee assistance program result is consistent with a previous study examining mental health training and attitudes toward support among a diverse national sample of Canadian PSP [21]. In Canada, the most common professional resource offered to PSP is the employee assistance program [21]; nevertheless, despite the broad access to these programs [33], most Canadian PSP (63.7%) reported they would never or only as a last resort access theirs [21]. The current results suggest that, despite PSP and CCG and C&P Officers having broad access to employee assistance programs, these programs are generally underutilized, highlighting the need to identify and address the barriers to accessing this resource. 

CCG and C&P officers with any mental health training indicated having access to all professional supports but reported higher willingness to access a physician (52.9%), chaplain (9.0%), and their employee assistance program (45.6%) than those with no training (prevalence of 49.2%, 6.8%, and 28.1%, respectively). Compared to those with no training, a higher proportion of CCG and C&P with any training indicated they would access support from a physician as an early resource (52.9%) and a smaller proportion indicated they would never access a psychologist (34.5%), a psychiatrist (39.5%), a chaplain (24.0%), or their employee assistance program (47.9%). The results suggest the type of professional CCG and C&P would access as an early resource varies regardless of training, but any training appears to increase willingness to access all professional support.

CCG and C&P with no mental health training indicated having access to all non-professional supports and were willing to access a spouse (70.6%), a friend (65.8%), a CCG or C&P colleague (21.5%), or leadership (10.7%). Accordingly, similar to willingness to access professional supports, willingness to access non-professional supports varied based on type. CCG and C&P with no mental health training indicated willingness to access support from a spouse (70.6%) or a friend (65.8%) as an early resource and would never or only as a last resort access a colleague (72.7%) or leadership (71.1%). The current results are consistent with previously surveyed Canadian PSP, wherein participants with no training reported they would never access a colleague (57.9%) or leadership (68.4%) [21]. 

CCG and C&P with any mental health training reported similar levels of perceived access to non-professional supports but reported higher willingness to access a spouse (73.8%), a colleague (25.4%), or leadership (11.2%) compared to participants with no mental health training. Comparable proportions of CCG and C&P officers with any mental health training indicated willingness to access support from a spouse (73.8%) and a friend (64.5%) as an early resource, but never or only as a last resort access support from a colleague (67.5%) or leadership (75.7%). Similar to professional supports, the results suggest the type of non-professional CCG and C&P would access as an early resource varies regardless of training, but any training appears to increase willingness to access all non-professional support. CCG and C&P willingness to access a spouse or a friend as an early resource suggests there may be significant benefits in providing mental health training for spouses, families, and friends of PSP.

The current results were consistent with results previously observed among other Canadian PSP with any mental health training; however, smaller proportions of other Canadian PSP reported they would never or only as a last resort access support from a colleague (51.2%) or leadership (66.8%) [21]. Irrespective of training, the type of non-professional support CCG and C&P would access as an early resource also varies and a larger proportion of CCG and C&P would never or only as a last resort access a colleague or leadership as compared to other Canadian PSP [21]. 

Regardless of receiving mental health training, most CCG and C&P indicated having access to all professional and non-professional supports, but CCG and C&P were most willing to access a physician, a spouse, or a friend. Similar results were observed among other Canadian PSP and further support that willingness to access support was more important than perceived access to support among PSP [21]. The current results suggest that the mental health training programs examined in the current study increased willingness to access both professional and non-professional supports compared to no training and suggest barriers to accessing specific supports as an early resource. Barriers to support should be identified and targeted by training as part of efforts to increase CCG and C&P willingness to access professional and non-professional support. Stigma is a well-documented barrier to seeking support for mental health among PSP [21,33,34,35,36,37,38]. The challenges from stigma are underscored by the unwillingness of CCG and C&P to access support from a colleague, a leader, or their employee assistance program. CCG and C&P officers may resist seeking care due to fear that others will deem them as weak or unreliable [33] or fear that seeking help for mental health challenges may impact progression in their careers (i.e., promotional opportunities [38,39]).

Informal mental health services may be perceived as more desirable than professional clinical services, particularly with respect to initial access to services [21,33]. This is supported by the unwillingness of CCG and C&P Officers to access a psychologist or a psychiatrist. Accessing support from a physician may be more discrete and raise less questions from colleagues or leaders than visiting a psychologist or psychiatrist. This is particularly important as one needs to be free of mental health concerns to go on ship, therefor having a mental health disorder could result in an inability to qualify for seafaring. PSP have also previously reported concerns that their choice to attend therapy would not remain confidential [40], thus leading to stigma and labeling by colleagues and leadership [33]. Additional mental health training and support for members and leaders targeting stigma towards those with mental health challenges is needed to increase willingness to access all types of professional and non-professional supports among CCG and C&P officers. 

Across all training types the proportion remained consistent for CCG and C&P indicating they would only access a physician, a spouse, or a friend as an early resource and would never access or only access as a last resort a psychologist, a psychiatrist, their employee assistance program, a chaplain, a colleague, and leadership. The current results suggest that all training categories provide similar core content or at least yield comparable results. This is possibly due to the programs being reactive in nature, except R2MR. The comparability of the included programs is further supported by the reported ratings of effectiveness for each training category. CCG and C&P officers reported each type of mental health training (i.e., CISM, CISM, Mental Health First Aid, Peer Support, R2MR) to be effective at improving their mental health, improving the mental health of their team members, reducing stigma, mitigating mental health injuries, increasing their knowledge about mental health, and helpful for responding to member of the public with mental health problems. CCG and C&P perceive the available training programs as effective at achieving the program objectives.

The current results indicated that prevalence of positive screens for PTSD, MDD, GAD, SAD, PD, and AUD was lower for those who received any mental health training (prevalence ranged from 7.5% to 22.4%) compared to those who received no training (prevalence ranged from 8.5% to 31.4%). The current results were consistent with results previously observed among other Canadian PSP [21]. There were only small differences in prevalence of screening positive for mental disorders (ranging 2.0% to 8.8%) across all training categories. Additionally, after adjusting for sociodemographic variables or other mental health training categories, no statistically significant associations were observed between training categories and decreased odds for positive screens for mental health disorders. Therefore, consistent with prior conclusions [13,21], the current results suggest that the different training categories likely yield comparable results and no one training program is superior to the others. Again, it is worth noting that most of the programs are reactive, and only one is proactive (i.e., R2MR). The current results also indicate that any mental health training may be beneficial for improving mental health challenges among CCG and C&P. This is further supported by the perceived effectiveness of each training program reported by CCG and C&P officers.

### Strengths and Limitations

The current study uses data provided by a national and diverse sample of CCG and C&P personnel; however, several limitations caveat the current results and provide directions for future research. First, the current sample, although demographically representative of CCG and C&P members, reflects approximately 6.15% of 6700 CCG and C&P members and includes larger proportions of CCG members (67.5%) than C&P members (26.0%) and members from British Columbia. Therefore, the current sample may not be entirely representative of the entire CCG and C&P workforce. Second, participation in the current study was anonymous, voluntary, and self-selected. Despite assurances of anonymity, stigmatizing attitudes about mental health may have inhibited some individuals from accessing the survey. Third, the collection method used an online survey, which may have impacted the number of participants. Many CCG and C&P members do not have easy access to computers or internet as they serve on ship, stations, or in the field and are often away for long periods of time. Despite being able to begin, leave, and return to the survey at their leisure, to ease survey response burden, not all participants completed all parts of the survey. As such, there is no way to know the average length of survey completion time or to understand why some participants did not complete the entire survey. 

Fourth, the screening measures for mental health disorders used in the current study are valid and reliable for use in clinical settings; nevertheless, diagnoses can only be made using clinical interviews with supporting collateral information [41]. A fully representative sample based on interview data would be beneficial but could deplete resources and be further impacted by stigma. Fifth, broad categories for each of the training options were used during data collection, therefore there may be important differences between training models that were overlooked. The overall comparability of the results suggests against large differences, but further research assessing the shared and unique content across the training categories is needed to make direct comparisons. The current study also only included a limited number out of many known mental health training programs, and the included programs were mostly reactive. Therefore, the results may be generalizable to reactive and not proactive programs, and other programs may yield more favorable results for PSP. Lastly, the current study examined only willingness to seek help and not actual help seeking behavior. Attitudes do not always correlate to actions and therefore conclusions cannot be made about the help-seeking behaviors of CCG and C&P officers [42]. Future research, both quantitative and qualitative, should explore treatment seeking behaviors among CCG and C&P Officers.

## 5. Conclusions

The current study assessed CCG and C&P attitudes towards accessing support from professional (i.e., physicians, psychologists, psychiatrists, employee assistance programs, chaplains) and non-professional (i.e., spouse, friend, colleagues, leadership) sources based on participating in different training program categories (i.e., CISM, CISD, Mental Health First Aid, peer support, R2MR) and associations with screening positive for mental health disorders. The current results provide potentially important information to inform decisions about implementing the mental health training programs included in the current study to improve attitudes toward and willingness to access supports among CCG and C&P officers. Regardless of having received mental health training, most CCG and C&P indicated having access to both professional and non-professional supports, but their willingness to access those supports varied. Training appears associated with higher perceptions of and willingness to access both professional and non-professional supports. Compared to those with no training, a higher proportion of CCG and C&P with any training indicated they would access support from a physician, a spouse, or a friend as an early resource, and a smaller or comparable proportion indicated they would never access a psychologist, a psychiatrist, a chaplain, their employee assistance program, a colleague, or leadership. A lower prevalence of positive screens for all mental disorders was observed for those with any training; but no significant associations were observed between mental health training categories and decreased odds for screening positive for mental disorders. The current results suggest that the mental health training programs examined in the current study are useful for improving attitudes toward support and willingness to access support and should be implemented among CCG, C&P, and other PSP groups to improve help-seeking attitudes and behaviors and reduce the impact of mental health challenges.

The current results suggest that the mental health training categories yield comparable results; however, it is worth noting that the included programs were primarily reactive. Further research is needed to assess the shared and unique content across each training program. Further research should also include more proactive or multimodal mental health training programs when assessing effectiveness or impact on willingness to access support. Perceptions of stigma and concerns about negative consequences may all be important factors influencing CCG and C&P willingness to access mental health supports. Additional research is needed to understand the rationale behind not accessing specific supports and how different mental health training programs may impact the support choices of CCG and C&P officers (i.e., peer support programs to increase willingness to access a colleague) The preference of CCG and C&P to access a spouse or a friend as an early resource suggests there may be significant benefits in providing mental health training for spouses, families, and friends of PSP. Additionally, the unwillingness of CCG and C&P to access their employee assistance program EAP, a colleague, or a leader suggests there may also be opportunities to improve such programs and provide members and leaders with new skills for reducing stigma and better support CCG and C&P mental health.

## Figures and Tables

**Table 1 ijerph-19-15734-t001:** Prevalence of mental health training categories and attitudes toward accessing professional mental health support.

	I Can and Would Access as an Early Resource % (*n*)	I Can Access but Would Never/Only as a Last Resort % (*n*)	I Don’t Have Access, but Would Access % (*n*)	I Don’t Have Access, but Would Never/Only as a Last Resort % (*n*)	I Don’t Know If I Have Access % (*n*)
Physician					
CISM	48.0(36)	41.3(31)	6.7(5)	^	^
CISD	58.1(18)	29.0(9)	^	^	^
Mental Health First Aid	54.2(45)	33.7(28)	7.2(6)	^	^
Peer Support	52.6(20)	34.2(13)	^	^	-
R2MR	58.2(57)	30.6(30)	7.1(7)	^	^
Any Training	52.9(90)	36.5(62)	7.6(13)	^	^
No Training	49.2(59)	43.3(52)	4.2(5)	-	^
Psychologist					
CISM	30.1(22)	39.7(29)	19.2(14)	^	6.8(5)
CISD	32.3(10)	35.5(11)	16.1(5)	^	^
Mental Health First Aid	33.7(28)	28.9(24)	22.9(19)	^	9.6(8)
Peer Support	26.3(10)	34.2(13)	26.3(10)	^	^
R2MR	35.4(34)	28.1(27)	19.8(19)	7.3(7)	9.4(9)
Any Training	30.4(51)	34.5(58)	19.0(32)	5.4(9)	10.7(18)
No Training	31.7(38)	42.5(51)	10.8(13)	5.8(7)	9.2(11)
Psychiatrist					
CISM	19.4(14)	47.2(34)	16.7(12)	^	11.1(8)
CISD	20.0(6)	40.0(12)	13.3(4)	^	16.7(5)
Mental Health First Aid	15.7(13)	36.1(30)	22.9(19)	9.6(8)	15.7(13)
Peer Support	13.2(5)	44.7(17)	18.4(7)	13.2(5)	^
R2MR	19.8(19)	33.3(32)	19.8(19)	10.4(10)	16.7(16)
Any Training	18.0(30)	39.5(66)	19.8(33)	7.8(13)	15.0(25)
No Training	23.7(28)	41.5(49)	11.0(13)	9.3(11)	14.4(17)
Employee Assistance Program					
CISM	39.2(29)	56.8(42)	^	-	^
CISD	54.8(17)	41.9(13)	-	-	^
Mental Health First Aid	51.8(43)	39.8(33)	^	^	^
Peer Support	36.8(14)	57.9(22)	^	-	^
R2MR	45.4(44)	49.5(48)	^	^	^
Any Training	45.6(77)	47.9(81)	^	^	3.6(6)
No Training	28.1(34)	65.3(79)	^	^	^
Chaplain					
CISM	16.4(12)	23.3(17)	-	35.6(26)	24.7(18)
CISD	20.0(6)	30.0(9)	-	26.7(8)	23.3(7)
Mental Health First Aid	15.7(13)	16.9(14)	^	43.4(36)	22.9(19)
Peer Support	21.1(8)	23.7(9)	^	34.2(13)	15.8(6)
R2MR	11.5(11)	18.8(18)	^	41.7(40)	26.0(25)
Any Training	9.0(15)	24.0(40)	^	40.1(67)	24.6(41)
No Training	6.8(8)	27.1(32)	^	43.2(51)	19.5(23)

Note. Training Categories are not mutually exclusive; CISM—Critical Incident Stress Management; CISD—Critical Incident Stress Debriefing; R2MR—Road to Mental Readiness. -: *n* = 0; ^: Sample size between 1 and 4, so data not presented.

**Table 2 ijerph-19-15734-t002:** Prevalence of mental health training categories and attitudes toward accessing non- professional mental health support.

	I Can and Would Access as an Early Resource % (*n*)	I Can Access but Would Never/Only as a Last Resort % (*n*)	I Don’t Have Access, but Would Access % (*n*)	I Don’t Have Access, but Would Never/Only as a Last Resort % (*n*)	I Don’t Know If I Have Access % (*n*)
Spouse					
CISM	75.7(53)	8.6(6)	^	^	8.6(6)
CISD	63.3(19)	^	^	^	^
Mental Health First Aid	75.0(60)	^	8.8(7)	6.3(5)	7.5(6)
Peer Support	64.9(24)	^	^	^	^
R2MR	71.7(66)	8.7(8)	5.4(5)	6.5(6)	7.6(7)
Any Training	73.8(121)	6.7(11)	6.1(10)	6.7(11)	6.7(11)
No Training	70.6(84)	4.2(5)	6.7(8)	9.2(11)	9.2(11)
Friend					
CISM	67.6(50)	27.0(20)	^	^	^
CISD	67.7(21)	25.8(8)	^	-	^
Mental Health First Aid	71.1(59)	26.5(22)	-	-	^
Peer Support	71.1(27)	23.9(9)	^	^	-
R2MR	63.9(62)	32.0(31)	^	^	^
Any Training	64.5(109)	30.2(51)	^	^	3.0(5)
No Training	65.8(79)	30.8(37)	^	^	^
CCG/DFO Colleague					
CISM	27.5(19)	68.9(51)	-	^	^
CISD	22.6(7)	77.4(24)	-	-	-
Mental Health First Aid	28.9(24)	66.3(55)	^	^	^
Peer Support	31.6(12)	60.5(23)	-	^	^
R2MR	24.7(24)	67.0(65)	^	^	^
Any Training	25.4(43)	67.5(114)	^	3.0(5)	3.0(5)
No Training	21.5(26)	72.7(88)	^	5.0(6)	-
CCG/DFO Leadership					
CISM	12.2(9)	75.7(56)	-	10.8(8)	^
CISD	^	83.9(26)	-	^	-
Mental Health First Aid	13.3(11)	77.1(64)	^	6.0(5)	^
Peer Support	^	81.6(31)	-	^	^
R2MR	11.3(11)	74.2(72)	^	9.3(9)	^
Any Training	11.2(19)	75.7(128)	^	8.9(15)	3.0(5)
No Training	10.7(13)	71.1(86)	-	9.9(12)	8.3(10)

Note. Training Categories are not mutually exclusive; CISM—Critical Incident Stress Management; CISD—Critical Incident Stress Debriefing; R2MR—Road to Mental Readiness. -: *n* = 0; ^: Sample size between 1 and 4, so data not presented.

**Table 3 ijerph-19-15734-t003:** Prevalence of screening positive for mental disorders among individuals who have had different mental health training categories.

Training Category	PTSD % (*n*)	MDD % (*n*)	GAD % (*n*)	SAD % (*n*)	PD % (*n*)	AUD % (*n*)
CISM	13.2(10)	22.4(17)	6.7(5)	17.3(13)	8.6(6)	10.0(7)
CISD	15.2(5)	^	^	15.6(5)	^	^
Mental Health First Aid	16.7(14)	20.2(17)	10.8(9)	16.7(14)	12.2(10)	8.0(6)
Peer Support	15.4(6)	15.4(6)	15.4(6)	17.9(7)	13.5(5)	^
R2MR	17.2(17)	24.2(24)	11.2(11)	18.2(18)	8.4(8)	8.8(8)
Any Training	16.7(29)	22.4(39)	12.2(21)	18.5(32)	8.4(14)	7.5(12)
No Training	18.0(22)	31.4(38)	24.0(29)	26.8(34)	8.5(10)	8.8(10)

Note. Training Categories are not mutually exclusive; AUD = Alcohol Use Disorder; CISM—Critical Incident Stress Management; CISD—Critical Incident Stress Debriefing; GAD = Generalized Anxiety Disorder; MDD = Major Depressive Disorder; PD = Panic Disorder; PTSD = Posttraumatic Stress Disorder; R2MR—Road to Mental Readiness; SAD = Social Anxiety Disorder. -: *n* = 0; ^: Sample size between 1 and 4, so data not presented.

**Table 4 ijerph-19-15734-t004:** Associations between screening positive for mental disorders among individuals who have had different mental health training categories.

Training Category	PTSD	MDD	GAD	SAD	PD	AUD
CISM						
OR (95% CI)	0.86(0.41,1.78)	0.86(0.47,1.57)	0.33(0.13,0.84) *	0.72(0.37,1.40)	0.94(0.37,2.38)	1.40(0.57,3.49)
AOR1 (95% CI)	0.74(0.31,1.72)	0.72(0.37,1.41)	0.39(0.14,1.09)	0.64(0.30,1.41)	1.06(0.36,3.17)	2.14(0.75,6.11)
AOR2 (95% CI)	0.67(0.23,1.93)	1.21(0.53,2.78)	0.32(0.08,1.19)	0.72(0.27,1.90)	0.62(0.14,2.81)	2.84(0.72,11.13)
CISD						
OR (95% CI)	1.04(0.39,2.82)	0.28(0.09,0.96) *	0.51(0.15,1.73)	0.66(0.24,1.77)	1.46(0.48,4.45)	0.79(0.18,3.50)
AOR1 (95% CI)	0.59(0.16,2.20)	0.32(0.09,1.12)	0.84(0.22,3.14)	0.88(0.26,2.91)	1.78(0.49,6.54)	1.39(0.26,7.37)
AOR2 (95% CI)	0.60(0.12,3.02)	0.53(0.12,2.38)	1.68(0.28,10.20)	1.30(0.27,6.21)	1.98(0.34,11.52)	1.27(0.17,9.66)
Mental Health First Aid						
OR (95% CI)	1.21(0.63,2.33)	0.73(0.41,1.32)	0.59(0.28,1.25)	0.68(0.36,1.28)	1.60(0.72,3.53)	1.02(0.39,2.63)
AOR1 (95% CI)	1.02(0.47,2.22)	0.56(0.29,1.11)	0.53(0.21,1.31)	0.54(0.25,1.17)	1.47(0.56,3.87)	1.01(0.36,2.86)
AOR2 (95% CI)	0.99(0.40,2.43)	0.64(0.30,1.38)	0.65(0.23,1.81)	0.57(0.23,1.37)	1.38(0.43,4.41)	0.67(0.18,2.48)
Peer Support						
OR (95% CI)	1.06(0.43,2.66)	0.53(0.21,1.31)	0.97(0.39,2.44)	0.78(0.33,1.85)	1.70(0.61,4.71)	0.64(0.15,2.82)
AOR1 (95% CI)	0.86(0.29,2.52)	0.34(0.12,0.97) *	1.04(0.36,3.01)	0.84(0.31,2.30)	1.46(0.40,5.29)	0.71(0.15,3.47)
AOR2 (95% CI)	1.09(0.30,3.89)	0.55(0.16,1.92)	2.00(0.50,7.99)	1.12(0.30,4.11)	1.25(0.22,6.96)	0.44(0.07,2.90)
R2MR						
OR (95% CI)	1.29(0.70,2.38)	0.98(0.57,1.67)	0.60(0.30,1.22)	0.76(0.42,1.36)	0.91(0.40,2.11)	1.18(0.49,2.81)
AOR1 (95% CI)	1.49(0.67,3.33)	0.85(0.43,1.66)	0.58(0.25,1.37)	0.78(0.36,1.67)	1.26(0.44,3.65)	1.48(0.48,4.56)
AOR2 (95% CI)	1.94(0.75,4.97)	1.34(0.61,2.93)	0.78(0.29,2.12)	1.06(0.44,2.55)	1.25(0.22,6.96)	1.50(0.37,6.08)
Any Training						
OR (95% CI)	1.32(0.76,2.28)	0.81(0.51,1.30)	0.60(0.34,1.07)	0.72(0.43,1.19)	0.89(0.43,1.84)	0.91(0.41,2.03)
AOR1 (95% CI)	1.20(0.61,2.35)	0.68(0.39,1.19)	0.64(0.32,1.26)	0.56(0.30,1.06)	1.01(0.42,2.42)	0.88(0.35,2.23)
AOR2 (95% CI)						

Training Categories are not mutually exclusive; AOR1 = adjusted odds ratio for sex, gender, age, education, ethnicity, marital status, province, and job category; AOR2 = adjusted odds ratio for the same variables as AOR1 in addition to all included categories of mental health training, which cannot be computed for any mental health training; AUD = Alcohol Use Disorder; CI = Confidence Interval; CISM—Critical Incident Stress Management; CISD—Critical Incident Stress Debriefing; GAD = Generalized Anxiety Disorder; MDD = Major Depressive Disorder; OR = unadjusted odds ratio; PD = Panic Disorder; PTSD = Posttraumatic Stress Disorder; R2MR—Road to Mental Readiness; SAD = Social Anxiety Disorder. * *p* < 0.05.

**Table 5 ijerph-19-15734-t005:** Prevalence of mental health training categories and attitudes toward the effectiveness of training.

	Responses % (*n*)	CISM % (*n*)	CISD % (*n*)	Mental Health First Aid % (*n*)	Peer Support % (*n*)	R2MR % (*n*)	Any Training % (*n*)
Improving your mental health	Yes	53.9(41)	66.7(22)	48.8(41)	61.5(24)	42.4(42)	42.5(74)
	Maybe	28.9(22)	18.2(6)	31.0(26)	30.8(12)	32.3(32)	35.6(62)
	No	17.1(13)	15.2(5)	20.2(17)	^	25.3(25)	21.8(38)
Improving the mental health of your team members	Yes	52.6(40)	60.6(20)	50.0(42)	64.1(25)	41.4(41)	39.7(69)
	Maybe	30.3(23)	27.3(9)	31.0(26)	25.6(10)	33.3(33)	37.9(66)
	No	17.1(13)	^	19.0(16)	^	25.3(25)	22.4(39)
Reducing stigma	Yes	63.2(48)	66.7(22)	64.3(54)	63.2(24)	55.6(55)	60.1(104)
	Maybe	23.7(18)	15.2(5)	22.6(19)	23.7(9)	25.3(25)	23.7(41)
	No	13.2(10)	18.2(6)	13.1(11)	13.2(5)	19.2(19)	16.2(28)
Mitigating Operational Stress Injuries	Yes	36.8(28)	42.4(14)	33.3(28)	46.2(18)	29.3(29)	27.0(47)
	Maybe	34.2(26)	39.4(13)	38.1(32)	35.9(14)	36.4(36)	39.7(69)
	No	28.9(22)	18.2(6)	28.6(24)	17.9(7)	34.3(34)	33.3(58)
Increasing your knowledge about mental health	Yes	77.6(59)	81.8(27)	75.0(63)	79.5(31)	71.7(71)	71.8(125)
	Maybe	18.4(14)	18.2(6)	17.9(15)	20.5(8)	19.2(19)	20.1(35)
	No	^	-	7.1(6)	-	9.1(9)	8.0(14)
Helping you to respond to members of the public with mental health problems	Yes	50.0(38)	69.7(23)	56.0(47)	61.5(24)	44.4(44)	41.4(72)
	Maybe	26.3(20)	21.2(7)	29.8(25)	30.8(12)	30.3(30)	33.3(58)
	No	23.7(18)	^	14.3(12)	^	25.3(25)	25.3(44)

Note. Training Categories are not mutually exclusive; CISM—Critical Incident Stress Management; CISD—Critical Incident Stress Debriefing; R2MR—Road to Mental Readiness. -: *n* = 0; ^: Sample size between 1 and 4, so data not presented.

## Data Availability

Not applicable.

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
