# Peer review of "Mental Health Training, Attitudes toward Support, and Screening Positive for Mental Disorders among Canadian Coast Guard and Conservation and Protection Officers"

_ijerph, 2022, doi:10.3390/ijerph192315734_

Round 1

Reviewer 1 Report

This article is devoted to a very relevant and practically significant topic. The article as a whole makes a good impression: it is prepared in accordance with the rules of modern scientific publications, it is well structured and formatted.

I would like to offer the authors some recommendations that may help improve the article:

1)     I propose to formulate the purpose of the study and add it to the abstract at the end of the Introduction

2)     In the Conclusion section, it is desirable to add practical recommendations based on the results obtained

3)     Perhaps the authors should consider a schematic representation of the main study findings to highlight the associations between the main study variables

4)     The text uses a lot of abbreviations, which makes it difficult to read. It might be worth replacing some of them with full terms and/or synonyms

I hope that the authors will be able to easily eliminate the shortcomings

Author Response

This article is devoted to a very relevant and practically significant topic. The article as a whole makes a good impression: it is prepared in accordance with the rules of modern scientific publications, it is well structured and formatted.

Thank you for your positive comments and feedback to strengthen the manuscript. We appreciate your time and effort spent reviewing the manuscript and hope you will find our updates satisfactory and the manuscript suitable for publication.

I would like to offer the authors some recommendations that may help improve the article:

  • I propose to formulate the purpose of the study and add it to the abstract at the end of the Introduction

Thank you for your comment and opportunity to strengthen the manuscript. We have clarified the purpose of the study in the introduction and added to the abstract.

  • In the Conclusion section, it is desirable to add practical recommendations based on the results obtained

Thank you for identifying this as an opportunity to strengthen the manuscript. We have added further implications and recommendations to the discussion and conclusion. Recommendations mentioned in the conclusion include: (1) providing any of the mental health training programs examined in the current research to improve attitudes toward support and willingness to access support and thus reduce the impact of mental health challenges; (2) further research to assess more proactive or multimodal mental health training programs and the shared and unique content across each training programs; (3) additional research to understand the rationale behind not accessing specific supports and how different mental health training programs may impact the support choices of PSP (i.e., peer support programs to increase willingness to access a colleague)

  • Perhaps the authors should consider a schematic representation of the main study findings to highlight the associations between the main study variables

Thank you for the suggestion. If the reviewer has specific findings they would like to see represented in a schema, we could work to add this. Otherwise pending an editorial request, we would opt out of adding a schematic representation due to the number of variables (i.e., mental health training programs, professional support, non-professional supports) included in the study, that may lead to an overly complicated diagram.

  • The text uses a lot of abbreviations, which makes it difficult to read. It might be worth replacing some of them with full terms and/or synonyms

Thank you for identifying this opportunity improve the presentation of the manuscript. We have removed some of the less common acronyms and reduced other acronyms that appear most frequently.

Reviewer 2 Report

The manuscript title Mental Health Training, Attitudes Toward Support, and Screening Positive for Mental Disorders Among Canadian Coast Guard and Conservation and Protection Officers Overall, the subject matter of this paper is modern, interesting, and useful to the general reader. The authors wrote it concisely, to the point, and well supported by the literature. However, it must be admitted that the content of the Research Findings is too short. The authors should consider expanding it so that the reader can see enough detail and understand it a bit more. These relevant research articles may also be considered referenced in order to link and expand to the general reader more clearly the condition of the issue. I suggest some issue for author response below.

- Web base self report survey in English and France languages, Do you have a way to check the language? How?

- How about researcher verify Mental Health Training program? It's an interesting topic. for the researchers to provide experiences to the target groups according to their needs. Appreciate.

- Reporting of comparative results between professional and non-professional groups In this research Is there a limitation or not?

Overall, it's a good article. modern and appropriately disseminated to the international mental health community

Author Response

The manuscript title “Mental Health Training, Attitudes Toward Support, and Screening Positive for Mental Disorders Among Canadian Coast Guard and Conservation and Protection Officers” Overall, the subject matter of this paper is modern, interesting, and useful to the general reader. The authors wrote it concisely, to the point, and well supported by the literature. However, it must be admitted that the content of the Research Findings is too short. The authors should consider expanding it so that the reader can see enough detail and understand it a bit more. These relevant research articles may also be considered referenced in order to link and expand to the general reader more clearly the condition of the issue. I suggest some issue for author response below.

Thank you for your positive comment and feedback to strengthen the manuscript. We appreciate your time and effort spent reviewing the manuscript and hope you will find our updates satisfactory and the manuscript suitable for publication.

We have added implications and recommendations to expand on the research findings.

- Web base self report survey in English and France languages, Do you have a way to check the language? How?

Thank you for this opportunity to clarify the methods in the manuscript. The survey was available in both English and French and participants were able to select which language to complete the survey at the start of the survey. All translations were completed in Qualtrics such that two versions of the survey existed, an English version and a French version.

- How about researcher verify Mental Health Training program? It's an interesting topic. for the researchers to provide experiences to the target groups according to their needs. Appreciate.

Thank you for your positive comment. In the conclusion we have a recommendation for future research to assess the unique and shared content of mental health training programs.

- Reporting of comparative results between professional and non-professional groups In this research Is there a limitation or not?

Thank you for the opportunity to clarify on this topic. We have selected to present separately professional, and non-professional supports to correspond with the questions the participants were asked in the survey. When presented with the list of professional supports, the participant could select all that apply. The same methods were used for non-professional supports. We believe the distinct is important because of the context the supports would be accessed in (i.e., at work, at home, in public). This also allows for the current results to be easily compared to previous research using the same methods.

Overall, it's a good article. modern and appropriately disseminated to the international mental health community.